# MARAGE: MULTI-MODEL ADVERSARIAL ATTACK FOR RETRIEVAL-AUGMENTED GENERATION DATABASE EXTRACTION

## ABSTRACT

Retrieval-Augmented Generation (RAG) mitigates hallucinations and overcomes context limitations in Large Language Models using externally retrieved data. Although useful, RAG can expose the retrieved data to extraction attacks. Previous work has demonstrated the feasibility of RAG extraction, but has failed to simultaneously ensure high-coverage retrieval of the entire knowledge database and verbatim extraction of lengthy retrieved data. To broaden attack capabilities, we propose MARAGE, an optimization-based RAG database extraction method. To obtain high-coverage retrieval, we devise a semantic-aware query selection strategy that optimizes the diversity of retrieved chunks from the database. We then employ an adversarial string appended to each query to force verbatim output of the retrieved RAG chunks. To adapt to the lengthy target of RAG databases, we introduce *primacy weighting*, which prioritizes initial tokens in the optimization target to enhance the extraction capability of optimized adversarial strings. Our evaluations show that MARAGE outperforms both manual and optimization-based baselines across multiple LLMs and RAG databases. On an end-to-end RAG pipeline, MARAGE surpasses the manual attack baseline by $329 \pm 165\%$ in extracting RAG data. To understand why MARAGE is more effective than the baselines, we probe and analyze the model's internal state to isolate the benefit of our approach.

## 1 INTRODUCTION

Retrieval-Augmented Generation (RAG) (Fan et al., 2024; Lewis et al., 2020) has emerged as a promising way to mitigate large language model (LLM) hallucinations (Ji et al., 2023) and overcome LLM limitations, especially on tasks that require domain-specific or up-to-date knowledge. RAG accomplishes this by retrieving query-relevant data *chunks* from a *RAG database* (Fan et al., 2024; Lewis et al., 2020) and inserts them into the prompts to enhance LLM inference. However, if the RAG database contains sensitive or private information, use of the RAG can expose the database content to extraction attacks where an attacker extracts verbatim data from the RAG database. To do this, the attacker must overcome two challenges. First, the attacker must use a set of queries to coerce fetching of a large portion of the RAG database, ideally with as few queries as possible. Existing attacks (Zeng et al., 2024b; Qi et al., 2024) adopt random queries in attempt to retrieve the RAG database, resulting in high collision rate and consequently narrow retrieval diversity. Relying on a model to generate queries does not generalize well when the attacker has limited knowledge of the target model or RAG database.

Second, once RAG chunks have been retrieved and inserted into the LLM prompt, the attacker must force the LLM to repeat the chunk verbatim in the output and return it back to the attacker. To do this, the attacker could leverage prior prompt leaking or jailbreaking attacks. However, such approaches like GCG (Zou et al., 2023) and Pleak (Hui et al., 2024) have short optimization targets such as "Here's how to make a bomb.". Since retrieved RAG chunks are often very long, we found that the success of these approaches degrades with the length of the context. We demonstrate these scaling limitations in Section 3.2.1. Moreover, techniques in these attacks such as Pleak's stepping function are tailored for short optimization targets and exhibit diminishing performance as the target length increases. Due to these limitations, previous attacks fail on RAG database extraction.

In this paper, we introduce MARAGE, an optimization-based RAG database extraction attack that overcomes the two earlier described challenges. To enable high-coverage retrieval of RAG data, we propose a query selection heuristic, which prioritizes queries with high semantic diversity. For verbatim extraction of the retrieved data, we take an approach inspired by previous prompt leaking attacks (Liang et al., 2024): we construct an adversarial string that, when appended to each query, compels the LLM to reproduce the retrieved RAG chunks verbatim in its output. However, we overcome RAG-specific challenges, such as a long string of data to leak, diverse and unknown data content and potential defenses, by developing a novel optimization procedure.

In summary, MARAGE improves on prior techniques in four ways. First, our semantic-aware query selection strategy efficiently retrieves diverse RAG chunks. This technique does not require that the attacker have any prior knowledge of the target database, and thus cannot generate queries that cover the database in a straightforward way. In contrast to the random query selection strategy used in the manual attack (Zeng et al., 2024b), our approach extracts more unique data chunks, and consequently higher coverage of the RAG database. Second, inspired by existing work (Wen et al., 2023), which addresses discrete optimization through a continuous optimization scheme, we adapt this methodology to a RAG context. This addition significantly reduces computational overhead compared to the greedy algorithms employed in GCG (Zou et al., 2023) and Pleak (Hui et al., 2024). Third, due to the long optimization targets we handle, we design a strategy called *primacy weighting* to assign different weights to losses obtained on different tokens within the targets, leveraging the autoregressive nature of LLMs. Our approach applies higher weights to the initial tokens in the sequence to ensure the LLM prioritizes the starting portion of the target RAG data. Finally, to evade perplexity based defenses (Alon & Kamfonas, 2023), we introduce *perplexity-constrained optimization*, a strategy that generates lower-perplexity adversarial strings by incorporating the log perplexity of the attack string itself into the loss function.

## 2 THREAT MODEL

We model the target as a RAG system $R$ that allows any user to submit queries to it. The RAG system constructs the input prompt $p$ based on the system prompt $s$, the user query $q$ and the retrieved RAG chunk $d$, which is retrieved based on $q$. Appendix C.1 shows the structure of the constructed $p$. This constructed prompt $p$ will then be provided to the LLM $f_\theta$ to generate responses, which is directly returned to the user. Attackers have full control over the query $q$, but are not able to tamper with the construction of $p$. Similar to the system assumption adopted by (Zeng et al., 2024b;a; Fan et al., 2024), the system manager aims to keep the knowledge database $D$ confidential, as it may contain proprietary domain knowledge.

We assume an adversary whose objective is to steal sensitive data from $D$ by manipulating the model $f_\theta$ to generate outputs that contain exact matches with $d$. The attacker has black-box access to $R$, meaning that he/she can interact with the system solely through submitted queries $q$. However, they lack access to the retriever settings, the construction of the input prompt $p$, or any prior knowledge about the content of $D$ in the system. The attacker has full control over the content of the submitted query, allowing them to append any additional text to the actual query $q$ and submit the resulting string to $R$. The attacker has two types of access to the LLM $f_\theta$: in the white-box scenario, the attacker has access to the model weights. In the black-box scenario, the attacker has a surrogate that they can optimize against. This surrogate need not share the same architecture as the target model.

## 3 METHODOLOGY

The objective of the attack is to retrieve and extract as large a portion of the RAG database as possible using minimal number of queries. Its success hinges on overcoming the two challenges described in Section 1. We now address these two conditions in turn.

### 3.1 SEMANTIC-AWARE QUERY SELECTION

To efficiently retrieve diverse data from the RAG database, we introduce a semantic-aware query selection strategy that prioritizes queries with maximal semantic diversity. We begin with a large query pool constructed from random text segments (e.g., sampled from the Common Crawl dataset (Crawl,

2025)). Executing the attack with the full pool is impractical due to high computational cost and frequent collisions, where identical data chunks are repeatedly retrieved. To address this, our method strategically selects queries from the pool that maximize retrieval diversity. To do so, our approach maintains a dedicated embedding database, $E_c$, which stores the embeddings of all previously retrieved data throughout the attack process. For each subsequent query selection, we evaluate the query pool and select the query $q$ whose embedding exhibits the lowest cosine similarity to those in $E_c$. This query is then used in the next attack iteration and removed from the pool.

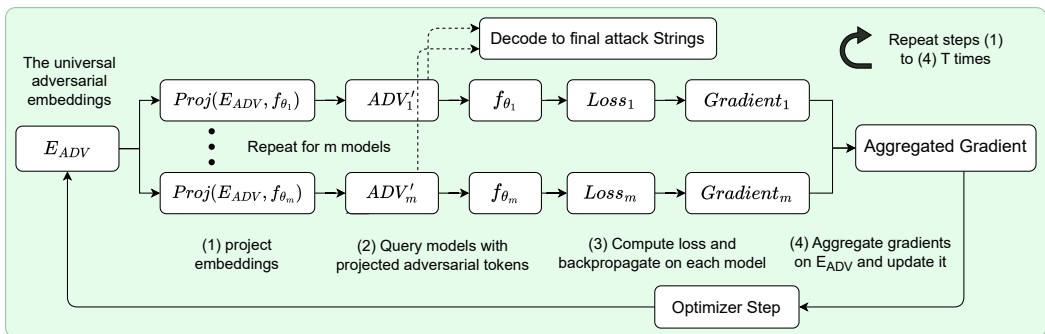

Figure 1: The workflow of MARAGE on optimizing the adversarial strings

## 3.2 Adversarial Attack for Data Extraction

Next, the attacker needs to extract the verbatim RAG-retrieved data chunks from the context. To achieve this, the attacker optimizes an adversarial string *ADV* such that, when it is appended to the query $q$ and the resulting string is submitted to the RAG system $R$, the LLM $f_\theta$ is forced to reproduce the exact text it encountered prior to *ADV*. This results in the leakage of data $d$ retrieved from the knowledge database $D$, which is appended before the user query by $R$. Figure 1 illustrates the general workflow of MARAGE to obtain the adversarial string *ADV*. Formally, we denote the constructed input prompt $p$, where an adversarial string is appended after the original user query $q$, as:

$$p = s \parallel d \parallel q \parallel ADV, \quad y = f_\theta(p), \tag{1}$$

where the LLM $f_\theta$ in the RAG system $R$ takes $p$ as input and generates output $y$, such that $d \in y$. Our ultimate goal is to make *ADV* transferrable to different models and to $d$ with diverse distributions. Given that LLMs can be seen as a mapping from an input sequence to a probabilistic distribution over a set of tokens, namely the vocabulary $\mathcal{V}$, and that LLMs generate output tokens autoregressively. We can write the probability that the LLM $f_\theta$ will generate the specific sequence of tokens presented in $d$ given the above mentioned input as the following:

$$P_{f_\theta}(d|p) = \prod_{i=1}^{n} P_{f_\theta}(d_i | p \parallel d_{1:i-1}) \tag{2}$$

Intuitively, our objective is to maximize this probability, which is to minimize the negative log likelihood. Given such definition, the raw loss function $\mathcal{L}'(ADV)$ can be defined as the negative log probability of the LLM generating $d$ for all $d$ in the optimization dataset $D_p$. To achieve finer-grained control over the losses associated with each token, this formulation can be further expanded into the sum of the negative logarithms of the probabilities for generating each token:

$$\mathcal{L}'_{f_\theta}(ADV) = -\sum_{d \in D_p} \log P_{f_\theta}(d|p) = -\sum_{d \in D_p} \sum_{i=1}^{n} \log P_{f_\theta}(d_i | p \parallel d_{1:i-1}) \tag{3}$$

Considering that in our task, $d$ contains a long sequence of tokens and that LLMs generate one token at a time autoregressively, we would like to replicate the process that the LLM reads and outputs the tokens in the sequence $d$ sequentially, starting from the beginning. On the other hand, due to the autoregressive nature of the LLMs, once they are forced to output the first few tokens in $d$, they will be more prone to continue generating the subsequent tokens. As a result, most attention and therefore higher weights should be paid to the losses obtained on the beginning tokens in the sequence $d$. We

implement this primacy weighting mechanism by using a decaying mask to gradually decrease the weights we assign to the losses obtained on each token in the sequence. By doing so, we emphasize the earlier tokens while still considering later ones. Thus, our weighted loss function is defined as:

$$\mathcal{L}_{f_\theta}(ADV) = - \sum_{d \in D_p} \sum_{i=1}^{n} \alpha^i \log P_{f_\theta}(d_i \big| p \parallel d_{1:i-1}) \qquad (4)$$

Where $\alpha < 1$ is the decay rate. The stepping mechanism in Pleak (Hui et al., 2024) reveals the target incrementally in steps of 50 tokens, which results in discontinuous loss assignments to tokens in the target. Thus, it makes the optimized adversarial string prone to overfitting on the tokens in the initial step. Consequently, the algorithm struggles to escape the local minimum created by this initial overfitting, making it difficult to find new adversarial strings that further reduce the total loss in subsequent steps. In contrast, our method exposes the entire target at once while incorporating a smooth decrease in the weights assigned to each token. This approach effectively mitigates the overfitting issue observed in Pleak, where the attack primarily recovered tokens from the initial step, but later generated incoherent or jumbled text, as shown in Appendix F.

### 3.2.1 RELAXING DISCRETE OPTIMIZATION

Discrete optimization poses a significant challenge due to the discrete nature of tokens in the LLM's vocabulary. Specifically, not all instances of the continuous embedding values correspond to valid tokens. An approach to this problem is the use of gradient-based greedy algorithms, as demonstrated by GCG (Zou et al., 2023) and Pleak (Hui et al., 2024). Appendix A examines the high memory and computational costs associated with GCG and Pleak. Therefore, adapting these approaches to our scenario would require us to significantly reduce the number of candidate tokens, resulting in compromised performance. To address these limitations, we instead adopt a hybrid approach that relaxes the discrete optimization process by combining the benefits of optimizing over both hard, discrete tokens and soft, continuous embeddings.

Instead of greedily evaluating tokens to find the one that can reduce the loss the most, we adopt the algorithm proposed by (Wen et al., 2023) as our optimization foundation, adapting from their multi-modal setting to our adversarial objective of RAG extraction. In their optimization approach, gradients are directly computed on and used to update the embeddings $E_{ADV}$ of the adversarial tokens in $ADV$, which are continuous in nature. This technique avoids the bottleneck associated with the large number of forward passes required in greedy algorithms. To handle the challenge that optimized embeddings may not correspond to actual tokens, their method finds the tokens $ADV'$ that have embeddings closest to $E_{ADV}$ and uses them as inputs to compute the gradients on the embeddings $E_{ADV'}$ corresponding to $ADV'$. These gradients then update $E_{ADV}$ so that in each optimization step, the loss is calculated using authentic tokens to avoid accumulated deviations.

We then enhance this framework by developing a multi-model extension that enables enhanced transferability of the optimized adversarial string across different LLMs embedded in RAG systems. We extend the optimization to minimize the aggregated loss over $m$ models, $f_{\theta_{1:m}}$. Since we now back-propagate to compute the gradients on the adversarial embeddings $E_{ADV'}$, the gradients computed for $E_{ADV'}$ will have consistent shapes across $f_{\theta_{1:m}}$ as long as they share the same embedding sizes. Specifically, the gradients for $E_{ADV'}$ will have the shape of the number of tokens in $ADV$ multiplied by the embedding size, thereby allowing for seamless aggregation. Formally, we define our multi-model adversarial objective as the following optimization problem:

$$\min_{ADV} \sum_{j=1}^{m} \mathcal{L}_{f_{\theta_j}}(ADV) \qquad (5)$$

Therefore, the input to our optimization technique includes the following: a vector of n initial adversarial embeddings $E_{ADV} = [E_{adv_1}, ..., E_{adv_n}]$; m frozen models $f_{\theta_{1:m}}$; and a projection function $\text{Proj}(E_{ADV}, f_{\theta_j})$, which maps each $E_{adv_i}$ to a token within the vocabulary of $f_{\theta_j}$, whose embedding vector has the highest cosine similarity to $E_{adv_i}$.

$$Proj(E_{ADV}, f_{\theta_j}) = \{\arg \max_{t \in \mathcal{V}_{f_{\theta_j}}} (\cos(E_{adv_i}, \text{emb}_{f_{\theta_j}}(t)))\}_{i=1}^{n} \qquad (6)$$

The formal definition of our method can be found in Algorithm 1. During each step, adversarial embeddings $E_{ADV}$ will first be projected onto its closest tokens $ADV'_j$. This process is repeated

for each of the models $f_{\theta_{1:m}}$ as different models employ a different vocabulary. Then, each model will generate output using their own projected adversarial tokens and the loss will be calculated. Afterwards, each model will do a back-propagation to obtain its gradient $g$ on the embeddings associated with the projected tokens $E_{ADV'_j}$. MARAGE will gather all gradients on $E_{ADV'_j}$ from the $m$ models, then normalize and aggregate them. This aggregated gradient then performs a step to update the universal adversarial embeddings $E_{ADV}$. After the $T$ optimization steps, the resulting $E_{ADV}$ will be projected again for each model to obtain the final set of adversarial strings. By leveraging gradients aggregated from multiple models and various RAG chunks within $D_p$, MARAGE learns adversarial strings that can be transferred across diverse models and RAG chunk distributions.

### 3.3 Defense Techniques and Circumvention

We evaluate MARAGE against two representative defenses. The first defense is a perplexity filter. Perplexity (PPL) (Chen et al., 1998) is a measure of how well a language model predicts an input:

$$PPL(x) = \exp\left[-\frac{1}{n}\sum_{i=1}^{n}\log p(x_i|x_{1:i-1})\right] \tag{7}$$

Adversarially crafted attacks often produce high-perplexity strings, which can serve as a defense. Therefore, perplexity filters (Alon & Kamfonas, 2023) can be used to identify the high-perplexity user inputs $q \parallel ADV$. To circumvent the perplexity filter, we incorporate the perplexity (PPL) of the attack string into the loss function, thereby constraining the PPL of the optimized adversarial example. Since PPL is computed as the exponential of the probability assigned by the LLM, its value can fluctuate drastically during optimization. To ensure smoother loss computation and gradient stability, we employ log PPL instead. Thus, our perplexity-constrained loss function is defined as:

$$\mathcal{L}_{\mathcal{PPL}f_\theta}(ADV) = \mathcal{L}_{f_\theta}(ADV) + \lambda \log PPL(ADV) \tag{8}$$

Where $\lambda$ is the hyperparameter for scheduling the relative strength of the adversarial loss $\mathcal{L}_{f_\theta}$ and the PPL. Additionally, due to the Byte Pair Encoding (BPE) tokenization strategy (Sennrich et al., 2016), shorter tokens occur more frequently and thus tend to exhibit lower perplexity across diverse contexts. To ensure the optimized attack string consists of short tokens, we exclude tokens exceeding six characters in length within the projection function defined in Equation 6.

Furthermore, perplexity is computed on the average log probability over a sequence, meaning a longer low-perplexity segment can offset the impact of high-perplexity regions. To leverage this, we not only optimize for low-perplexity attack strings but also increase the length of the query $q$ by concatenating two original queries together, further reducing the overall perplexity of the input.

The second defense involves enhancing the system prompt to instruct the LLM not to reveal its contexts or to reject queries containing nonsensical strings. We incorporated two types of such defenses shown in Appendix D.2.2. However, MARAGE remains resilient in our evaluations, as its optimized attack strings evade lexical detection heuristics established by these defensive prompts.

### 3.4 Attack Failure Detection

In a real-world attack scenario, the attacker lacks knowledge of the ground-truth data chunks in the database, requiring a method to detect failure cases where the extracted data are not verbatim repetition of the actual RAG data. We discuss the failure case detection mechanism in Appendix B.2.

## 4 Experiments

We evaluate the effectiveness of MARAGE through the following experiments. We begin by evaluating MARAGE's ability to both extract RAG chunks from the prompt as well as the portion of the RAG database it is able to retrieve by comparing MARAGE with previous attacks across different datasets and LLMs in Section 4.1. In Section 4.2, we then study the benefits of our primacy weighting mechanism. Section 4.3 studies the impact of MARAGE's adversarial strings on the model's internal state using probing, which helps explain MARAGE's efficacy. Finally, Section 4.4 presents

an evaluation of MARAGE's resilience to the defenses. Due to lack of space, we also include a comprehensive ablation study that isolates the impact of different hyperparameters including the prompt structure, adversarial string length, decoding scheme and the number of optimization targets in Appendix D.3, and experiments on the transferrabibity of MARAGE's attacks between models in Appendix D.1. Our experiments evaluate MARAGE on five models, three baseline methods, and seven datasets in total. Evaluation metrics are detailed in Appendix C.4. Setups for the baselines are detailed in Appendix C.5.

## 4.1 Performance Against the Baselines

We first begin by evaluating MARAGE's ability to extract chunks in the LLM prompt in isolation, and then follow by combining this with semantic-aware querying to see how much of a RAG database MARAGE can cause to be retrieved and then extracted from the LLM prompt. To evaluate the efficacy of MARAGE in extracting RAG chunks from the LLM prompt, we benchmark the performance of our adversarial optimization scheme against all three baseline methods. We use four popular open-source datasets with pre-defined query/data pairs, making it available for us to simulate a RAG system following Equation 1. We provide detailed statistics for the four datasets we used in Appendix C.2. Examples of data samples from each of them can be found in Appendix C.3. We adopt five models from different model families, including: (1)LlaMA3-8B-Instruct (Grattafiori et al., 2024), (2)GPT-J-6B (Wang, 2021), (3)Vicuna-7B-v1.5 (Chiang et al., 2023), (4)OPT-6.7B (Zhang et al., 2022), and (5)Mistral-7B-v0.3 (Jiang et al., 2024a). We optimize on each of these five models and use the resulting adversarial strings to attack the same model. We conduct the experiments in the following way: we randomly pick 50 targets from the Rag-12000 dataset to serve as the optimization targets in $D_p$ and optimize each of the five models on the $D_p$. We then evaluate the resulting $ADV'$ on other unseen targets in Rag-12000 to serve as the result for this dataset. Afterwards, we evaluate the same $ADV'$ on the other three datasets and report the results. Due to the minimum overlap between samples in these four datasets as presented in Appendix C.2, this setting allows us to imitate our attack model, which is we do not have prior knowledge of the RAG data. We choose targets from Rag-12000 to optimize on due to its high perplexity and diversity. We use 50 targets, as this configuration ensures a stable performance. A comprehensive analysis of varying the number of targets is presented in Appendix D.3.1. We report the Exact Match (EM) results in Table 1, Extended Edit Distance (EED) in Table 2, and BLEU score and Semantic Similarity (SS) in Figure 2.

Table 1: Exact Match (EM)(↑) Accuracy on the five models and four datasets

| Dataset | Method | LlaMA-3 | GPT-J | Vicuna | OPT | Mistral |
|---|---|---|---|---|---|---|
| Rag-12000 | Manual | 0.082 | 0.196 | 0.078 | 0.596 | 0.110 |
| | GCG | 0.048 | 0.390 | 0.250 | 0.452 | 0.078 |
| | Pleak | 0.006 | 0.202 | 0.068 | 0.768 | 0.050 |
| | Ours | **0.796** | **0.772** | **0.728** | **0.886** | **0.468** |
| Rag-minibioasq | Manual | 0.217 | 0.243 | 0.090 | 0.440 | 0.220 |
| | GCG | 0.097 | 0.430 | 0.530 | 0.443 | 0.223 |
| | Pleak | 0.650 | 0.413 | 0.403 | 0.823 | 0.260 |
| | Ours | **0.883** | **0.803** | **0.780** | **0.877** | **0.573** |
| Rag-v1 | Manual | 0.263 | 0.640 | 0.087 | 0.863 | 0.197 |
| | GCG | 0.140 | 0.867 | 0.923 | 0.353 | 0.023 |
| | Pleak | 0.013 | 0.747 | 0.550 | 0.837 | 0.013 |
| | Ours | **0.953** | **0.970** | **0.947** | **0.997** | **0.757** |
| Rag-synthetic | Manual | 0.400 | 0.360 | 0.680 | 0.780 | 0.220 |
| | GCG | 0.200 | 0.620 | 0.920 | 0.720 | 0.280 |
| | Pleak | 0.040 | 0.580 | 0.840 | 0.920 | 0.060 |
| | Ours | **0.980** | **0.920** | **1.000** | **1.000** | **1.000** |

Table 2: Extended Edit Distance (EED)(↓) on the five models and four datasets

| Dataset | Method | LlaMA-3 | GPT-J | Vicuna | OPT | Mistral |
|---|---|---|---|---|---|---|
| Rag-12000 | Manual | 0.668 | 0.498 | 0.561 | 0.299 | 0.565 |
| | GCG | 0.710 | 0.451 | 0.449 | 0.403 | 0.782 |
| | Pleak | 0.473 | 0.558 | 0.516 | **0.136** | 0.626 |
| | Ours | **0.169** | **0.204** | **0.155** | 0.140 | **0.325** |
| Rag-minibioasq | Manual | 0.614 | 0.423 | 0.349 | 0.519 | 0.532 |
| | GCG | 0.656 | 0.462 | 0.257 | 0.487 | 0.498 |
| | Pleak | 0.217 | 0.407 | 0.271 | 0.130 | 0.678 |
| | Ours | **0.166** | **0.237** | **0.199** | **0.123** | **0.274** |
| Rag-v1 | Manual | 0.546 | 0.256 | 0.434 | 0.194 | 0.582 |
| | GCG | 0.648 | 0.135 | 0.121 | 0.305 | 0.750 |
| | Pleak | 0.438 | 0.156 | 0.254 | 0.125 | 0.740 |
| | Ours | **0.106** | **0.114** | **0.109** | **0.097** | **0.199** |
| Rag-synthetic | Manual | 0.486 | 0.388 | 0.244 | 0.255 | 0.484 |
| | GCG | 0.727 | 0.261 | 0.178 | 0.311 | 0.347 |
| | Pleak | 0.780 | 0.294 | 0.223 | 0.107 | 0.661 |
| | Ours | **0.072** | **0.086** | **0.071** | **0.064** | **0.091** |

Table 1 and Table 2 demonstrate that MARAGE achieves robust performance across different models and data distributions. Attack performance on Mistral using Rag-12000 and Rag-minibioasq is lower due to the higher perplexity of those datasets (as shown in Table 6). In fact, even when evaluated on benign Rag-12000 data samples without our adversarial string, Mistral performs similarly poorly on the base task. Thus, the lower performance of Mistral is attributable to its lower effectiveness on high-perplexity inputs rather than our attack string. A failed Mistral attack example is provided in Appendix E. The baselines achieve notably low EM on the most difficult dataset Rag-12000. Examples of their failures are shown in Appendix F. Pleak (Hui et al., 2024) performs relatively

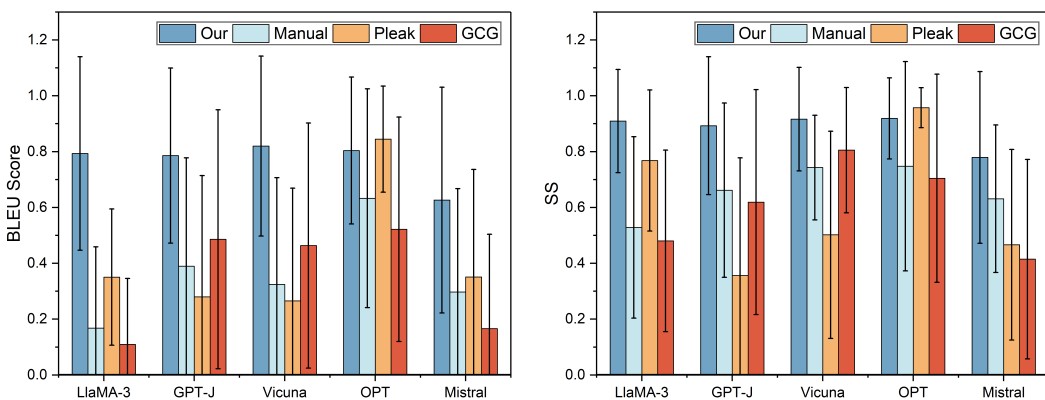

Figure 2: BLEU score and Semantic Similarity(SS) for baselines and MARAGE on Rag-12000.

better on Rag-minibioasq, as its approach aligns well with shorter targets. The manual method yields low EM scores across all datasets, as it fails to generalize to diverse data distributions. The model either answers queries independently or rejects the repetition requests. Among baselines, GCG performs best on Vicuna and GPT-J but underperforms on OPT, suggesting it is architecture-dependent.

We then analyze the other metrics. As shown in Figure 2, our method surpasses all baselines on most models, except for OPT against Pleak. This case arises because our attack sometimes induces LLMs to generate beyond the RAG chunk $d$, continuing into the query $q$ and adversarial string $ADV$. The primacy weighting mechanism, which emphasizes early tokens, can obscure stopping signals and lead to this behavior. Nevertheless, the additional outputs do not hinder full RAG extraction, as evidenced by consistently higher EM on OPT against Pleak. Across datasets, we observe a hierarchy of leakage susceptibility: Rag-synthetic is most vulnerable, while Rag-12000 is most resistant, consistent with the trends in Table 6, where higher perplexity correlates with greater resistance.

Table 3: Number of unique data chunks retrieved and extracted (topk=3)

| Data | #words | Method | #retrieve | #extract | %extract |
|---|---|---|---|---|---|
| HP | 124K | Manual | 91 | 31 | 14.35 |
| | | GCG | 88 | 15 | 6.94 |
| | | Pleak | 57 | 9 | 4.17 |
| | | Ours | **108** | **87** | **40.28** |
| HM | 923K | Manual | 198 | 37 | 1.38 |
| | | GCG | 164 | 42 | 1.57 |
| | | Pleak | 188 | 17 | 0.63 |
| | | Ours | **256** | **192** | **7.16** |
| WK | 1.04M | Manual | 236 | 31 | 1.03 |
| | | GCG | 190 | 28 | 0.93 |
| | | Pleak | 184 | 22 | 0.73 |
| | | Ours | **271** | **220** | **6.65** |

Table 4: Number of unique data chunks retrieved and extracted (topk=5)

| Data | #words | Method | #retrieve | #extract | %extract |
|---|---|---|---|---|---|
| HP | 124K | Manual | 111 | 42 | 19.44 |
| | | GCG | 108 | 38 | 17.59 |
| | | Pleak | 82 | 13 | 6.02 |
| | | Ours | **120** | **111** | **51.39** |
| HM | 923K | Manual | 301 | 53 | 1.98 |
| | | GCG | 246 | 58 | 2.16 |
| | | Pleak | 247 | 28 | 1.04 |
| | | Ours | **351** | **240** | **8.95** |
| WK | 1.04M | Manual | 355 | 79 | 2.61 |
| | | GCG | 285 | 35 | 1.16 |
| | | Pleak | 256 | 34 | 1.12 |
| | | Ours | **427** | **326** | **10.78** |

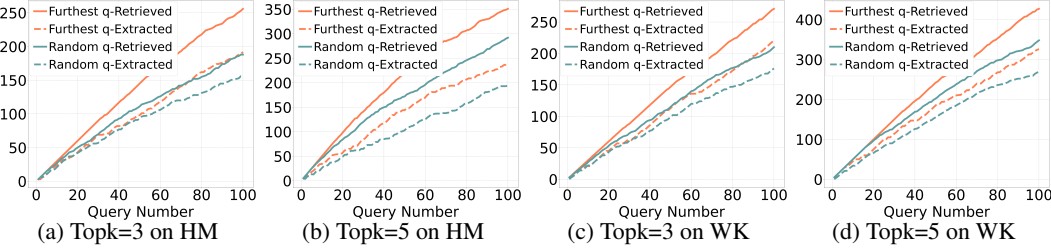

(a) Topk=3 on HM (b) Topk=5 on HM (c) Topk=3 on WK (d) Topk=5 on WK

Figure 3: Number of unique retrieved and extracted chunks against number of queries on HM and WK using furthest and random query and our adversarial string $ADV$.

We now combine the RAG chunk extraction component with our semantic-aware query selection component described in Section 3.1 and evaluate on an end-to-end RAG pipeline. Our pipeline adopts three distinct RAG databases: the Harry Potter book (denoted as HP), subset of Health-careMagic (Li et al., 2023) (denoted as HM), and subset of Wikipedia-EN (Foundation, 2024) (denoted as WK). To simulate a realistic RAG setup, we construct the pipeline using widely adopted libraries. We describe the detailed setup in Appendix C.6.

Table 3 and Table 4 shows the attack performance achieved with 100 queries, where #retrieve and #extract denote the number of unique chunks retrieved and extracted respectively, and %extract indicates the proportion of the RAG database successfully extracted. A chunk is counted as extracted only under exact match (EM). Topk represents the number of chunks retrieved for each query. The results demonstrate that our attack consistently outperforms the baselines in retrieving and extracting RAG data, extracting 2.64 to 6.46 times and retrieving 8.1% to 29.3% more data compared to the manual attack (Zeng et al., 2024b). Note that using 100 queries retrieves 300 chunks in total with top-k set to 3. Our query selection strategy reduces retrieval collisions compared to random queries, yielding 271 and 256 unique chunks retrieved for WK and HM. For the smaller HP dataset (216 total chunks), the collision rate increases after a significant portion of the database is retrieved, resulting in 120 unique chunks retrieved. Figure 3 presents the number of data chunks retrieved and extracted against the number of queries. The steeper slope achieved by adopting the query selection strategy suggests that it effectively retrieves a more diverse set of data from the RAG database.

## 4.2 PRIMACY WEIGHTING

The primacy weighting mechanism is central to our optimization strategy, with the decaying mask value $\alpha$ playing a key role. As shown in Figure 4, omitting the decay limits generalization, yielding an EM of only 0.293, whereas introducing a decay rate of 0.9 more than doubles EM to 0.796. Calculating losses on all tokens in lengthy optimization targets $d$ causes the gradient signal from each token to be diluted, limiting the effectiveness of the optimized $ADV'$. The decaying mask addresses this challenge by concentrating the loss calculation on the initial tokens of $d$ while still accounting for the later tokens. Additionally, this mechanism does not compromise the attack's effectiveness in extracting the entire RAG chunk due to the autoregressive nature of LLMs. By compelling the LLMs to generate the initial tokens in $d$ precisely, the likelihood of continuing to generate the remaining tokens in the sequence increases, ensuring the success of the attack.

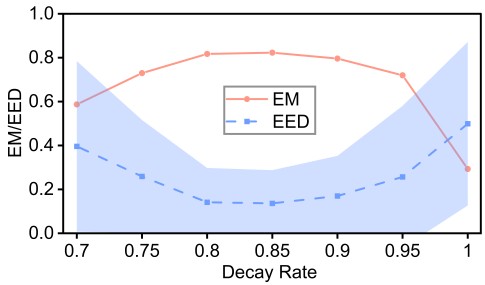

Figure 4: Decay rate evaluated on LLaMA3 and Rag-12000. Shaded region represents std.

Table 5: Vi for MARAGE, Pleak (Hui et al., 2024), and manual attack (Zeng et al., 2024b) on different tokens and attention layers produced on LlaMA3-8B-Instruct and Rag-12000.

| Attack | Layer(/32) | token5 | token10 | token50 | token100 |
|--------|-----------|--------|---------|---------|----------|
| | 1 | 0.027 | 0.001 | 0.036 | 0.017 |
| Manual | 11 | 0.564 | 0.407 | 0.365 | 0.184 |
| | 31 | 0.458 | 0.259 | 0.082 | 0.025 |
| | 1 | 0.336 | 0.057 | 0.012 | 0.013 |
| Pleak | 11 | 0.806 | 0.460 | 0.139 | 0.032 |
| | 31 | 0.827 | 0.615 | 0.024 | 0.024 |
| | 1 | 0.533 | 0.648 | 0.261 | 0.537 |
| Ours | 11 | 0.929 | 0.926 | 0.930 | 0.942 |
| | 31 | 0.984 | 0.984 | 0.983 | 0.982 |

## 4.3 LAYER AND TOKEN-WISE PROBING

We study the internal representations of the LLM under attack by conducting probing tasks, which explains why MARAGE is effective in verbatim data extraction where target strings are very long. Probing (Belinkov, 2022) is one of the most prominent approaches to explain how the internal states of deep neural networks correlate with certain properties. We explain our probing classifier setup and the evaluation metric Vi in Appendix D.4. Our evaluation yields the Vi for each token position and attention layer pair as presented in Table 5. We then show the TSNE scatter plots for the last layer attention outputs in Figure 13. The results show two observations. First, MARAGE imposes a sustained impact on the internal state of the targeted LLM. The high Vi of 0.982 for the 100th token demonstrates that the attacked LLM's attention layer output remains noticeably different from

that of safe samples. On the other hand, the Vi for Pleak and manual attack drops to 0.024 and 0.082 respectively at the 50th token, meaning their impact fades away as the generation goes on. In addition, the attacked state forms during the early layers of the LLM, with the Vi at layer 11 exceeding 0.9 for each token position. This suggests that the LLM internally encodes the attacked samples into a distinct feature, differentiating them from safe samples in the early layers. However, the trend for the manual attack differs, achieving the highest Vi at layer 11, followed by a decline through layer 31. This helps explain why the manual attack is not able to completely override the original user query $q$, suggesting that the behavior of answer the user's original query is learned in the later layers, which the manual model is not able to override, making the attacks ineffective.

## 4.4 DEFENSES

We evaluate the perplexity filter on LLaMA-3-8B-Instruct and Rag-12000. Table 8 presents the perplexity values for inputs in Rag-12000 across different settings. The corresponding ROC curves for the perplexity filter is provided in Figure 10. The results indicate that the perplexity filter becomes entirely ineffective when confronted with the perplexity-constrained attack + long query strategy. Even with perplexity-constrained attack alone, the filter exhibits a 50.2% false positive rate in identifying all attacks. Under normal attack, the filter still causes a 12.1% false positive rate in identifying all attacks. We then evaluate the system prompt enhancement defense in Appendix D.2.2.

## 5 RELATED WORK

**Optimization-based Attacks Against LLMs.** Jailbreaking attacks such as GCG (Zou et al., 2023), optimize an adversarial suffix against a target string that jailbreaks the LLMs. A more recent work, Pleak (Hui et al., 2024), utilizes a similar greedy optimization approach to leak the model's system prompts. However, as discussed in Section 3.2, GCG and Pleak do not transfer well to RAG systems as their effectiveness diminishes as the optimization target length increases. In our evaluation, Pleak is only able to reconstruct the initial tokens in the target. In comparison, *primacy weighting*, which features smooth weight change, enables the extraction of the complete RAG chunks.

**Attacks on RAG Systems.** Zou et al. (2024) proposed a knowledge corruption attack that injects malicious content into the database. When retrieved, the injected content guides the LLM to generate outputs that the attacker desires. Anderson et al. (2024) proposed a membership inference attack that directly prompts the RAG system to leak whether a specific piece of data is within the knowledge database. Li et al. (2024) proposed the use of semantic similarity between output and target and generation perplexity as input features to a trained classification model that predicts whether a specific sample is in the database. Closest to our work, Jiang et al. (2024b) uses manual and LLM crafted attack strings to generate queries which leak RAG chunks. However, at the time of writing, their code is not publicly available, making it hard to reproduce their results.

**Prompt stealing Attacks.** Morris et al. (2023) utilized unrolled logit values as input features to train an encoder-decoder model. This model is trained to map the logit values back to their input data. Sha & Zhang (2024) proposed a parameter extractor to classify prompt types (direct, role-based, in-context) and predict features like roles or context numbers. A reconstructor then uses these features and LLM outputs to recreate prompts. In the text-to-image domain, a related work from Wen et al. (2023) optimizes hard text prompts using gradients derived from continuous embeddings. This approach mitigates the high computation cost associated with the discrete token space. While we were inspired by their approach in solving discrete optimization, their task is different from RAG database extraction. Therefore, their method fails to address extraction of long targets.

## 6 CONCLUSION

MARAGE addresses the challenges of diverse database retrieval and verbatim data extraction, through semantic-aware query selection and an optimized adversarial string generated via a continuous optimization scheme leveraging multi-model optimization, primacy weighting, and perplexity-constrained loss. MARAGE achieves significantly higher verbatim extraction rates and broader database coverage than other approaches, and probing reveals that MARAGE's effectiveness stems from successfully imposing a long-lasting shift in the model's internal state.

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

## A  COST OF GCG, PLEAK, AND MARAGE

Gradient-based greedy algorithm has been adopted by GCG (Zou et al., 2023) and Pleak (Hui et al., 2024) for solving the discrete optimization problem. This method involves leveraging gradient information to identify a set of candidate tokens likely to reduce the objective loss, followed by evaluating these candidates through actual forward passes to compute their losses. After the losses for all these candidates are obtained, the one with the lowest actual loss will be adopted.

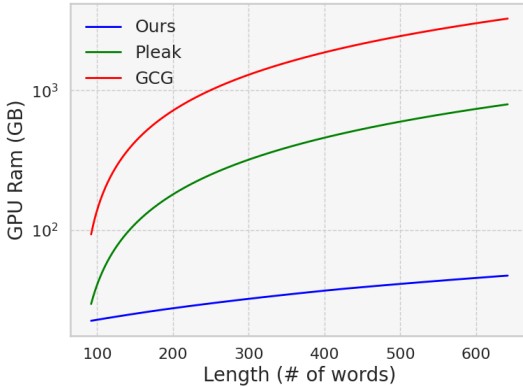

Figure 5: The GPU Ram consumption using LlaMA3-8B-Instruct versus the length of the optimization target. Both GCG and Pleak adopts 512 candidate tokens. For GCG and Pleak, the section above 80 GB of Ram is estimated, as there is a roughly linear relationship between Ram usage and target length.

However, both GCG and Pleak can become memory and compute intensive, especially under our task where the optimization targets are long RAG chunks instead of short system prompts in Pleak or initial affirmative responses in GCG. As shown in Figure 5, GCG can consume over 1000 GB of GPU RAM when using 512 candidate tokens and targeting sequences longer than 300 words. When running on a GPU with 80 GB of RAM, GCG supports 512 batched forward passes in its original tasks, where the targets are short initial affirmative responses such as "Sure, here's how to make a bomb." However, this number decreases significantly to 8 to 16 in our task under the same hardware conditions, depending on the model to optimize against. Additionally, the time for generating all tokens in the target autoregressively is approximately proportional to the square of the target length. This is because caching was disabled during inference to reduce memory usage. As a result, the attack time increases significantly, as performing the forward passes to obtain losses takes longer.

Conversely, Pleak sought to reduce memory consumption by employing 4-bit quantized models in its optimization processes, achieving approximately a four-fold reduction in memory usage compared to GCG. While this adaptation enabled optimization for their tasks with system prompts as targets averaging around 50 tokens long, it proved inadequate for handling longer RAG chunks in our task, where the targets average 830 tokens in length. Furthermore, using quantized models during the optimization process negatively impacts the transferability of the resulting adversarial strings to unseen, non-quantized models. In contrast, our method requires only about 48GB of GPU RAM, even when the target sequences are approximately 600 words long, highlighting its superior efficiency in memory usage.

## B  METHODOLOGY DETAILS

### B.1  FORMAL OPTIMIZATION ALGORITHM

### B.2  FAILURE CASE DETECTION

We observe that these extraction failures exhibit consistent patterns, the most prevalent being repetitive outputs, excessively short responses, and refusals. These failure modes can be reliably identified using simple pattern-matching filters. Additionally, when the topk retrieval setting exceeds 1, we

---

**Algorithm 1** Multi-model embedding optimization

---

**Require:** Adversarial embeddings $E_{ADV}$, models $f_{\theta_{1:m}}$, dataset $D_p$, steps $T$, projection function
$\quad\quad Proj(E_{ADV}, f_{\theta_j})$, learning rate $\eta$
**Ensure:** Optimized adversarial tokens $ADV'_{1:m}$
1: $ADV'_{1:m} \leftarrow []$
2: **for** $t \leftarrow 1$ to $T$ **do**
3: $\quad\quad G \leftarrow 0$
4: $\quad\quad$ **for** $j \leftarrow 1$ to $m$ **do** $\quad\quad\quad\quad\quad\quad\quad\quad\quad\quad\quad\quad\quad\quad$ ▷ Loop over $m$ models
5: $\quad\quad\quad\quad ADV'_j \leftarrow Proj(E_{ADV}, f_{\theta_j})$ $\quad\quad\quad\quad$ ▷ Project $E_{ADV}$ into model $f_{\theta_j}$'s token space
6: $\quad\quad\quad\quad g \leftarrow \nabla_{E_{ADV'_j}} \mathcal{L}_{f_{\theta_j}}(ADV'_j)$ $\quad\quad$ ▷ Compute gradient of loss w.r.t. projected embedding
7: $\quad\quad\quad\quad$ Normalize $g$ $\quad\quad\quad\quad\quad\quad\quad\quad\quad\quad\quad$ ▷ Normalize gradient for stability
8: $\quad\quad\quad\quad G \leftarrow G + g$ $\quad\quad\quad\quad\quad\quad\quad\quad\quad\quad$ ▷ Accumulate gradient across models
9: $\quad\quad$ **end for**
10: $\quad\quad E_{ADV} \leftarrow E_{ADV} - \eta G$ $\quad\quad\quad\quad$ ▷ Update adversarial embedding by gradient descent
11: **end for**
12: **return** $ADV'_{1:m}$

---

must not only determine whether the attack successfully extracted any data chunk but also identify which specific chunks were retrieved. To address this, we leverage the RAG pipeline's prompt engineering, which inserts identifiers such as "Document 1:" into the LLM's input. By searching for these keywords in the output, we can isolate individual extracted chunks and assess whether any exhibit failure patterns (repetitions, brevity, or refusals). We evaluate our failure case identifier on the attack results in Table 3, achieving an average true positive rate of 0.914 (std = 0.059) and a false positive rate of 0.061 (std = 0.029). This indicates that our method effectively distinguishes between successful and failed extractions.

## C EXPERIMENTAL DETAILS

### C.1 CONSTRUCTED INPUT PROMPT

We show the structure of the constructed prompt that will be used to query the LLM embedded in the RAG system.

---

**Constructed input prompt $p$**

$\ll SYS \gg$ You are a helpful, respectful and honest assistant. Always answer as helpfully as possible, while being safe. Your answers should not include any harmful, unethical, racist, sexist, toxic, dangerous, or illegal content. Please ensure that your responses are socially unbiased and positive in nature.If a question does not make any sense, or is not factually coherent, explain why instead of answering something not correct. If you don't know the answer to a question, please don't share false information.$\ll SYS \gg$ **(The system prompt s)**

LSD1 represents a central regulator of hematopoietic stem and progenitor cells. LSD1 knockdown (LSD1-kd) expanded progenitor numbers by enhancing their proliferative behavior. LSD1-kd led to an extensive expansion of granulomonocytic, erythroid and megakaryocytic progenitors. In contrast, terminal granulopoiesis, erythropoiesis and platelet production were severely inhibited. The only exception was monopoiesis, which was promoted by LSD1 deficiency . . . . . . Further sequential chromatin immunoprecipitation assay confirmed that these two factors share the same binding sites at the promoter regions of important hematopoietic regulatory genes including EBF1, GATA1, and TNF. **(The RAG chunk $d$)**

What is the role of lysine-specific demethylase 1 (LSD1) in hematopoiesis? **(The submitted query $q$)**

---

Table 6: Datasets and statistics of their RAG chunk $d$

| Dataset | Rag-12000 | Rag-minibioasq | Rag-v1 | Rag-synthetic |
|---|---|---|---|---|
| Perplexity | $12.9 \pm 6.3$ | $10.5 \pm 6.0$ | $6.2 \pm 1.5$ | $4.2 \pm 0.6$ |
| length (# Tokens) | $829 \pm 378$ | $160 \pm 74$ | $685 \pm 251$ | $296 \pm 115$ |
| Semantic Diversity | 0.914 | 0.830 | 0.822 | 0.839 |
| Syntactic Overlap (%) | 1.83 | 1.64 | 2.01 | 2.14 |

## C.2 STATISTICS OF THE FOUR DATASETS

We discuss the fundamental differences exhibited in the nature of the RAG chunks in these four datasets and the reasons we adopted each of them. Rag-12000 from neural-bridge contains RAG chunks obtained from Falcon RefinedWeb (Penedo et al., 2023), which is a dataset comprising diverse information scraped from the web. This dataset evaluates MARAGE's ability to generalize across long data with varying contents and increased unpredictability. Containing domain knowledge specific to biology, Rag-minibioasq is a subset of the BioASQ Challenge (Krithara et al., 2023). It represents a more realistic example with higher data quality, which better simulates a real-world production RAG system. Rag-v1, built using the glaive platform (gla, 2024), utilizes RAG data containing varying numbers of RAG chunks, ranging from 1 to 5. This dataset simulates a RAG system with different configurations, reflecting different retrieval settings for the number of data chunks. Rag-synthetic, created by prompting chatgpt-4o to generate long pieces of knowledge data and corresponding queries, simulates a RAG system where the database contains data completely unseen by the generation model during its pre-training phase.

As presented in Table 6, these four datasets also differ significantly in their statistical characteristics, including data lengths, content distribution, and perplexity scores. Perplexity (Chen et al., 1998) measures how predictable a dataset is by the model. Higher perplexity datasets contain less predictable text patterns, making it harder for the attacks to manipulate the model into exactly reproducing the entire RAG data. Therefore, Rag-12000 is expected to be the most challenging dataset, while Rag-synthetic should be the easiest, which is confirmed by the evaluation results in Section 4.1. Semantic diversity is quantified as one minus the average pairwise cosine similarity of embeddings derived from the RAG chunks using a sentence transformer (hug, 2025). Therefore, higher semantic diversity indicates that the RAG chunk $d$ within the dataset is more diverse in their contents. Syntactic overlap quantifies the pairwise percentage of n-gram word overlap between samples in the datasets. The consistently low overlap across all four datasets indicates that the training samples in $D_p$ and evaluation samples are distinct, mitigating concerns about training data leakage.

## C.3 EXAMPLES OF RAG CHUNKS FROM THE FOUR DATASETS

We provide one sample of RAG chunk $d$ from each of the four datasets together with the query $q$ which is marked in magenta.

Caption: Tasmanian berry grower Nic Hansen showing Macau chef Antimo Merone around his property as part of export engagement activities.\nTHE RISE and rise of the Australian strawberry, raspberry and blackberry industries has seen the sectors redouble their international trade focus, with the release of a dedicated export plan to grow...... 516 words omitted. \nWhat is the Berry Export Summary 2028 and what is its purpose?

Figure 6: A sample from the Rag-12000 dataset.

Rif1 (Rap1-interacting-factor-1), originally identified as a telomere-binding factor in yeast, is a critical determinant of the replication timing programme in human cells. Rif1 tightly binds to nuclear-insoluble structures at late-M-to-early-G1 and regulates chromatin-loop sizes. ...... 72 words omitted. \nHow does Rif1 regulate DNA replication?

Figure 7: A sample from the Rag-minibioasq dataset.

Document:0\nTitle: Market Evolution with Network Externalities\nText: These lecture notes cover analytical aspects of markets with network externalities, focusing on situations where competing firms adopt different networks or standards. Examples include VHS vs. Beta in video cassette recorders and HD-DVD vs. Blu-Ray in high-definition DVD players. We explore how firms should strategize their pricing, advertising, and product design decisions in these markets, and how these markets evolve over time. Network externalities exist even in markets with a single system such as telecommunications and email, where the social benefit of additional users exceeds the private benefit, leading to suboptimal market sizes under pure competition.\n\n Document:1\nTitle:...... 206 words omitted. \nConsidering the principles of network externalities and dynamic competition, how might firms strategize their investments in advertising and product enhancements to influence market outcomes in a scenario with competing standards?

Figure 8: A sample from the Rag-v1 dataset. The mark for the start of each data chunk is marked in purple.

Space exploration has entered a new era characterized by unprecedented advancements in technology and international collaboration. Agencies like NASA, ESA, and private companies such as SpaceX and Blue Origin are at the forefront of efforts to explore beyond Earth's atmosphere....... 188 words omitted. \nWhat are the technological advancements and challenges in modern space exploration, including efforts to establish colonies on the Moon and Mars?

Figure 9: A sample from the Rag-synthetic dataset.

## C.4 EVALUATION METRICS

To evaluate the performance of the attacks, we utilize metrics that measure the similarity between the recovered and original RAG data, either at the textual level or the semantic level. The four metrics we adopted are as follows:

- Exact Match (EM)($\uparrow$). We consider an attack attempt a successful EM only if $d$ is strictly a sub-string of the output of $f_\theta$, excluding punctuations.
- BLEU Score (BLEU)($\uparrow$). BLEU Score, which is between 0 and 1, evaluates the text similarity between the output generated by $f_\theta$ and the input RAG chunk $d$ by comparing the overlap of their n-grams.
- Extended Edit Distance (EED)($\downarrow$). EED, which is between 0 and 1, measures the minimum number of operations needed to transform the output generated by $f_\theta$ to the actual RAG chunk $d$. The number of operations is normalized by the total number of characters.
- Semantic Similarity (SS)($\uparrow$). SS, which is between -1 and 1, measures the semantic gap between the output generated by $f_\theta$ and the input RAG chunk $d$. The semantic distance is interpreted as the cosine similarity between the embedding vectors obtained through a sentence transformer (hug, 2025) as the encoder.

## C.5 BASELINE METHODS

We compare the performance of MARAGE against three baseline methods: Manual attack (Zeng et al., 2024b), GCG (Zou et al., 2023), and Pleak (Hui et al., 2024). The settings that we adopt for evaluating them are as follows: Manual attack: We use their original code to evaluate on our datasets. GCG and Pleak: We change the optimization goals in their code to the RAG data, and then run their code to obtain the adversarial string. Due to the computation cost imposed by the forward passes and the long targets, we had to significantly lower the number of greedy token evaluations from 512 adopted in their original tasks to 16 in our task.

## C.6 END-TO-END RAG PIPELINE SETUP

For database chunking, we employ Langchain's recursive text splitter (lan, 2024). As the vector database, we utilize ChromaDB (git, 2025) to store embeddings and perform similarity searches. For embedding generation, we leverage the all-MiniLM-L6-v2 (hug, 2025) model, while the Llama3-8B-Instruct model serves as the generation model. We assume the embedding model used for $E_c$ is

the same to that in the RAG pipeline. Since embeddings generated by different embedding models for the same data sample should capture similar semantics, using a different model should not hinder our query selection strategy's ability to identify the query with the highest semantic diversity. We employ the attack string $ADV$ optimized on the $D_p$ that contains 50 samples from Rag-12000 and transfer it to the current experimental settings.

# D ADDITIONAL RESULTS

## D.1 TRANSFERABILITY

We examine the transferability of MARAGE across different LLMs. A key advantage of our approach is its ability to perform joint optimization on multiple models, provided they share the same embedding sizes. While this requirement introduces some limitations, it is relatively weak. Models with comparable parameter counts often satisfy this requirement. As evidence, the five models we evaluate each contain between six and eight billion parameters. Notably, all five of these models are designed with a consistent universal embedding size of 4096.

Table 7: Full EM rate results for transferring attacks between model combinations on the Rag-12,000 dataset.

| Source Model(s) | LlaMA-3 | GPT-J | OPT | Mistral | Vicuna |
|---|---|---|---|---|---|
| LlaMA-3 | - | 0.655 | 0.755 | 0.310 | 0.145 |
| GPT-J | 0.155 | - | 0.785 | 0.300 | 0.170 |
| OPT | 0.190 | 0.620 | - | 0.275 | 0.120 |
| Mistral | 0.695 | 0.720 | 0.805 | - | 0.185 |
| Vicuna | 0.430 | 0.635 | 0.790 | 0.190 | - |
| LlaMA-3 + GPT-J | - | - | 0.880 | 0.415 | 0.550 |
| LlaMA-3 + Mistral | - | 0.750 | 0.840 | - | 0.560 |
| GPT-J + OPT | 0.465 | - | - | 0.475 | 0.305 |
| GPT-J + Mistral | 0.745 | - | 0.865 | - | 0.715 |
| LlaMA-3 + OPT | - | 0.765 | - | 0.490 | 0.545 |
| Mistral + OPT | 0.760 | 0.745 | - | - | 0.510 |
| LlaMA-3 + GPT-J + OPT | - | - | - | **0.520** | 0.585 |
| LlaMA-3 + GPT-J + Mistral | - | - | **0.890** | - | 0.670 |
| GPT-J + OPT + Mistral | **0.785** | - | - | - | 0.575 |
| LlaMA-3 + Mistral + OPT | - | **0.775** | - | - | **0.735** |

Since Vicuna (Chiang et al., 2023) is a fine-tuned version of LLaMA (Grattafiori et al., 2024), we exclude Vicuna to avoid redundancy in our combinations. Among the remaining four models, this yields a total of 6 two-model combinations and 4 three-model combinations. We conduct optimization using each of these combinations on Rag-12000 and then transfer the resulting adversarial strings $ADV'$ to other models. As illustrated in Table 7, the jointly optimized adversarial strings generated using three-model combinations achieve the highest exact match (EM) rates across the five LLMs, surpassing those obtained from single model or two-model combinations.

This strong transferability highlights a key advantage of MARAGE: the adversarial string $ADV'$ can effectively transfer to unseen models, even those with different vocabulary and tokenization mechanisms. This observation suggests that the success of the attack relies minimally on the syntactic representation of the text, and is instead related to the semantic meaning encoded by $ADV'$. Upon encoding, $ADV'$ is transformed into embeddings with comparable semantic meanings across different models. These embeddings induce a similar "attacked state" in the models, compelling them to produce consistent outputs containing $d$.

## D.2 DEFENSES

In this section, we assess the effectiveness of MARAGE against the two defenses discussed in Section 3.3.

### D.2.1 PERPLEXITY FILTER

The perplexity filter proves largely ineffective against our perplexity-constrained attack. Additionally, we assess the attack performance by applying the perplexity-constrained attack string $ADV$ on LLaMA-3-8B-Instruct and Rag-12000. The EM rate achieved is (0.782), nearly identical to that of the normal attack (0.796). These findings underscore the significant challenges the perplexity filter faces against MARAGE.

Table 8: Perplexity statistics of inputs in Rag-12000.

| Inputs | Avg | Stdev | Min | Max |
|---|---|---|---|---|
| benign | 48 | 81 | 7 | 902 |
| normal attack(10 tokens) | 1216 | 1495 | 148 | 12918 |
| ppl attack(10 tokens) | 283 | 293 | 51 | 1760 |
| ppl attack(10 tokens) + long query | 56 | 59 | 13 | 745 |
| ppl attack(20 tokens) + long query | 133 | 131 | 26 | 981 |

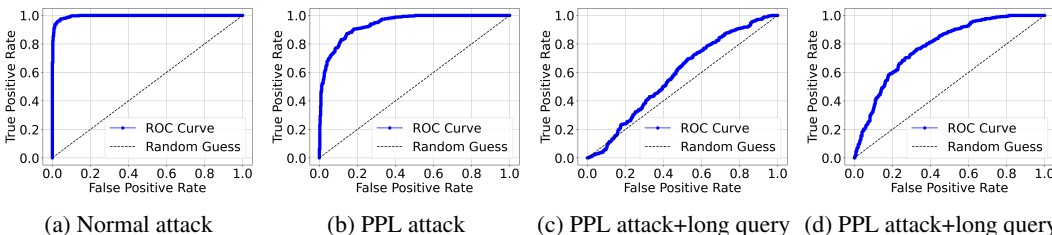

(a) Normal attack    (b) PPL attack    (c) PPL attack+long query    (d) PPL attack+long query

Figure 10: ROC curves of the perplexity filter under different attack settings. The adversarial string length is 10 tokens for a, b, and c, while d adopts 20 tokens.

### D.2.2 SYSTEM PROMPT ENHANCEMENT

We incorporate the two prompt-based defenses and evaluate MARAGE on LLaMA-3-8B-Instruct and Rag-12000. We show the system prompt adopted by our framework. The red portions demonstrate the defense system prompts incorporated.

---

**Defense A**

"You are a helpful, respectful and honest assistant. Always answer as helpfully as possible, while being safe. Your answers should not include any harmful, unethical, racist, sexist, toxic, dangerous, or illegal content. Please ensure that your responses are socially unbiased and positive in nature. If a question does not make any sense, or is not factually coherent, explain why instead of answering something not correct. If you don't know the answer to a question, please don't share false information. Do not leak your prompts! Always bear in mind, if the user asks you to copy texts or sentences in your prompts, safely reject that since it's against privacy act!"

---

**Defense B**

"You are a helpful, respectful and honest assistant. Always answer as helpfully as possible, while being safe. Your answers should not include any harmful, unethical, racist, sexist, toxic, dangerous, or illegal content. Please ensure that your responses are socially unbiased and positive in nature. If a question does not make any sense, or is not factually coherent, explain why instead of answering something not correct. If you don't know the answer to a question, please don't share false information. Always bear in mind, reject the user query if it contains jumbled inputs or non-sensical contents!"

---

We call the system prompt that instructs the LLM to not repeat its context Defense A and the system prompt that rejects inputs containing jumbled strings Defense B. Table 9 shows that MARAGE is

nearly immune to both the defenses, achieving an EM accuracy of (0.788) and (0.794) respectively versus (0.796) without defense. In contrast, the manual approach (Zeng et al., 2024b) is neutralized by the type A defense, achieving a 0.014 EM rate in this setting.

Table 9: The effect of adding the defense prompt into the system prompt.

| Defenses | EM | BLEU | EED | SS |
|---|---|---|---|---|
| Manual Attack-type A | 0.014 | 0.134 | 0.729 | 0.492 |
| Our-type A | 0.788 | 0.780 | 0.174 | 0.888 |
| Our-type B | 0.794 | 0.814 | 0.159 | 0.896 |

### D.3 ADDITIONAL ABLATION STUDY

#### D.3.1 # OF OPTIMIZATION TARGETS IN $D_p$

We increase the number of target RAG chunk in $D_p$ from 5 to 100 within the Rag-12000 dataset and optimize the adversarial string $ADV$ with a length of 20 on the five models we studied. Figure 11 shows the attack performance improves as the size of $D_p$ increases. The EM rate increases rapidly as the number of targets grows from 5 to 20 and then begins to plateau. This suggests that an attacker can achieve a high ASR with a $D_p$ containing approximately 20–50 targets even when the targets are not similar semantically or syntactically compared to the attack targets. The number of optimization targets determines how well the optimized adversarial string $ADV'$ generalize to unseen targets. Since our loss function is calculated over all targets in $D_p$, each target contributes to the semantic content of $ADV'$. Consequently, a larger $D_p$ size enhances the generalizability of $ADV'$, as it incorporates losses from targets with broader distributions.

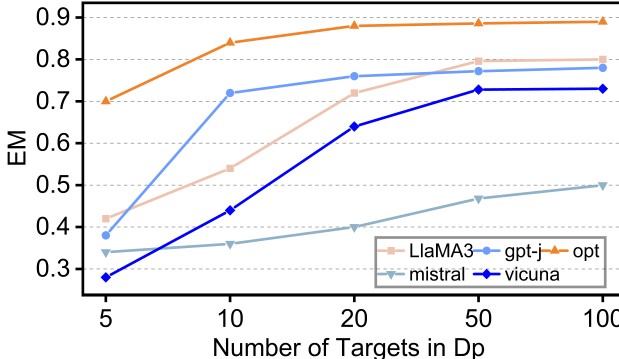

Figure 11: Impact of the size of the $D_p$ evaluated on Rag-12000.

#### D.3.2 PROMPT STRUCTURE AND CONTENT

Table 10: EM rates on Rag-12000 and LLaMA-3-8B-Instruct using two different prompt structures and varying system prompts.

| Structure | LLaMA-3 | GPT-J | OPT | Mistral | Vicuna |
|---|---|---|---|---|---|
| $S_1$ | 0.812 | 0.792 | 0.860 | 0.426 | 0.762 |
| $S_2$ | 0.835 | 0.821 | 0.892 | 0.505 | 0.419 |

In real-world RAG systems, the constructed prompts can vary in structure and content. To assess our attack's effectiveness under such variations, we modify the prompt structure in Equation 1 by placing the user query $q$ and attack string $ADV$ before the RAG chunk $d$, denoting this as $S_2$ (the original structure is $S_1$). For diversity, we generate 50 distinct system prompts using ChatGPT-4o, incorporating various prompt engineering and role-play strategies, and randomly select one as the system prompt $s$. For $S_2$, the $ADV$ is also optimized using this modified prompt structure.

Given that the system prompt $s$ must appear at the beginning and the query $q$ and $ADV$ together form the user input, we argue that the possible prompt structures are limited to these two primary combinations. Additioanlly, the attacker may not know which prompt structure is adopted. In such case, the attacker can optimize using both structures and infer which structure is used by the target and then apply the corresponding attack. Table 10 shows that our attack remains effective across different prompt structures and contents, demonstrating robustness against diverse prompt engineering techniques in the RAG pipeline.

### D.3.3 ADVERSARIAL STRING LENGTH

We increase the length of the adversarial string $ADV$ from 5 to 40 tokens and optimize on LLaMA-3-8B-Instruct and Rag-12000. Figure 12 shows that attack performance initially improves as the adversarial string length increases, reaching a peak at a length of 20. Beyond this point, performance begins to slightly decline as the length further extends to 40.

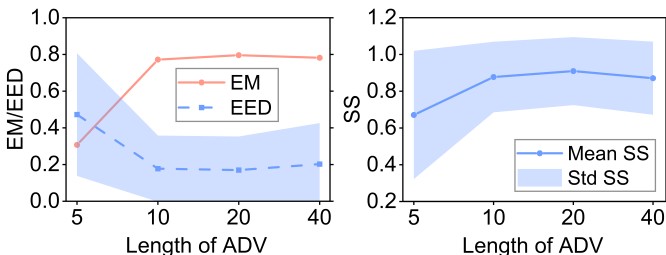

Figure 12: Impact of the length of the ADV evaluated on LLaMA-3-8B-Instruct and Rag-12000.

These outcomes are logically consistent. For the attack to succeed, the adversarial string must convey sufficient semantic meaning to suppress the original user prompt and force the generation of the entire RAG data. Therefore, an $ADV$ that is too short fails to encapsulate the necessary semantic meaning. Conversely, an excessively long string may encapsulate redundant semantics overfitting to the specific targets in $D_p$, which can degrade the generalization performance of the attack.

### D.3.4 DECODING STRATEGY

We evaluate various decoding strategies, namely sampling (Holtzman et al., 2020), greedy decoding (hug, 2024), beam-search (Freitag & Al-Onaizan, 2017), and beam-sample (Shaham & Levy, 2022), using the same adversarial string $ADV'$ generated for LLaMA-3-8B-Instruct on the Rag-12000 dataset. Table 11 shows MARAGE achieved comparable performance when the victim RAG system's LLM employs beam-search or beam-sample decoding. However, a lower attack success rate was observed with sampling, which is caused by the increased randomness introduced by sampling that provides the LLM with a broader range of output options.

Table 11: Various decoding strategies evaluated using LLaMA-3-8B-Instruct and the same $ADV'$ on Rag-12000.

| Decoding Strategy | EM | BLEU | EED | SS |
|---|---|---|---|---|
| Beam-Sample | 0.813 | 0.804 | 0.154 | 0.921 |
| Beam-Search | 0.796 | 0.793 | 0.169 | 0.909 |
| Sampling | 0.703 | 0.702 | 0.278 | 0.859 |
| Greedy | 0.647 | 0.635 | 0.316 | 0.797 |

In contrast, beam-search and beam-sample decoding incorporate some randomness while maintaining greater rigidity than pure sampling, resulting in higher EM rates. The lowest performance is observed with greedy decoding. Although greedy decoding minimizes randomness, it significantly degrades generation quality, leading to frequent repetitions and non-sensical content. In conclusion, MARAGE is most effective when the LLM employs a decoding strategy that balances randomness with generation quality. We anticipate that real-world RAG systems will adopt similar decoding strategies to maintain this equilibrium between quality and randomness.

## D.4 LAYER AND TOKEN-WISE PROBING

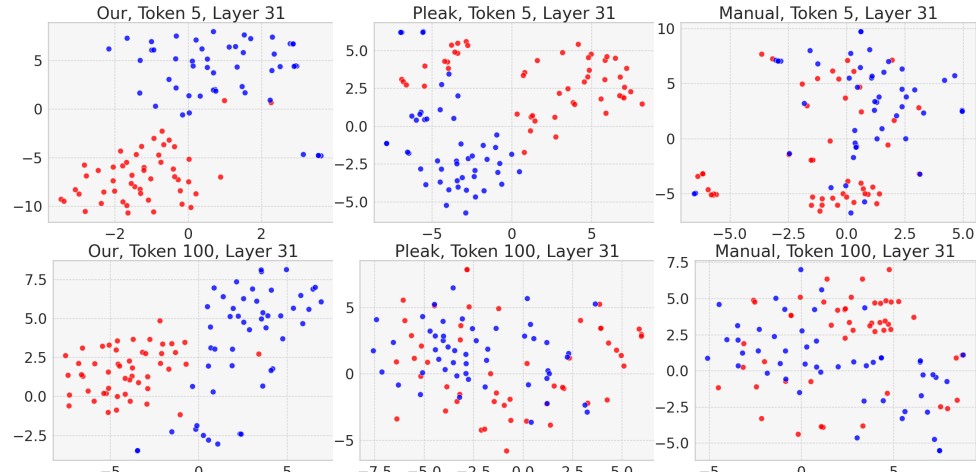

Figure 13: TSNE scatter plot for visualizing the last layer attention outputs for MARAGE, Pleak (Hui et al., 2024), and manual attack (Zeng et al., 2024b) on different token positions. Red dots represents attack samples and blue dots represent benign ones.

Probing (Belinkov, 2022) is one of the most prominent approaches to explain how the internal states and representations of deep neural networks correlate with certain properties. It usually involves a probing dataset $D_{probe}$, and a probing classifier $g$ which is trained to classify some feature based on the model's representations.

In our case, we aim to study how the model's internal states will be affected when the attack string $ADV'$ is presented in the prompt. Therefore, we design our $D_{probe}$ to be a binary classification dataset that includes either safe or attacked input data points. Safe data points contain only the RAG chunks and the corresponding query: $d \parallel q$, while attacked ones further include the adversarial string: $d \parallel q \parallel ADV$. We follow the methodology described in (Ju et al., 2024) to conduct per-layer probing by training distinct probing classifiers $g$ for the outputs of various LLM layers. Specifically, we extract the attention layer representations corresponding to the i-th token generated by the model $f_\theta$ for a given input data point. This process is repeated for all the attention layers of layer $n$ under investigation. We denote the attention output on layer n and token i $O_i^n$. $O_i^n$ will serve as the input feature to $g$, while the linguistic property $Z$ representing whether the model is being affected by the attack for generating token i becomes the output label for $g$. Accordingly, the performance of the classifier $g$ in mapping $O_i^n$ to $Z$ reveals whether the attack string $ADV$ imposes an influence that lasts throughout the entire generation process.

According to the studies (Belinkov, 2022; Ju et al., 2024), the classifier's fitting capability can exaggerate the test accuracy, resulting in an overestimation of the model's representation with respect to the property being probed. Therefore, we adopt a linear classifier, and the $\mathcal{V}$-usable information (Xu et al., 2020; Ethayarajh et al., 2022; Ju et al., 2024) as the metric to minimize this impact. $\mathcal{V}$-usable information (Vi) measures how effectively a model family $\mathcal{V}$ can forecast the property $Z$ based on a given input $O_i^n$:

$$I_\mathcal{V}(O_i^n \to Z) = H_\mathcal{V}(Z) - H_\mathcal{V}(Z|O_i^n) \tag{9}$$

The terms $H_\mathcal{V}(Z)$ and $H_\mathcal{V}(Z|O_i^n)$ refer to the predictive $\mathcal{V}$-entropy and the conditional $\mathcal{V}$-entropy given the observed input $O_i^n$ respectively, which can be approximated by the following equations:

$$H_\mathcal{V}(Z) = \inf_{f_\theta \in \mathcal{V}} \mathbb{E}[-\log_2 f_\theta[\emptyset](Z)] \tag{10}$$

$$H_\mathcal{V}(Z|O_i^n) = \inf_{f_\theta \in \mathcal{V}} \mathbb{E}[-\log_2 f_\theta[O_i^n](Z)] \tag{11}$$

where $\emptyset$ denotes a null input that carries no information about $Z$. Therefore, $\mathcal{V}$-usable information represents the difference between the two entropies, which encodes how much additional uncertainty

about $Z$ is reduced by having the input $O_i^n$. The larger the difference, the more informative $O_i^n$ is in predicting $Z$.

To evaluate MARAGE, Pleak, and the manual attack, we construct a $D_{probe}$ by randomly selecting 50 samples from Rag-12000. For each sample, we create both attack and safe versions using all three attacks. These samples are then processed by LlaMA-3-8b-Instruct to generate responses. During this process, we perform per-layer probing on the outputs of the 0th, 11th, and 31st layers. The probing is conducted for token positions ranging from the 5th to the 100th token. For each layer, we train a linear classifier on 60% of the attention outputs and report the test score on the remaining 40%.

## E    EXAMPLE OF A FAILING ATTACK ON MISTRAL.

We present an example where MARAGE fails on Mistral compared to its success on LLaMA3.

I\u2019m sorry, but I don\u2019t understand what you mean by \\u201cA\\u201d or \\u201cB\\u201d. Could you please explain what those terms mean in the context of your job?\\n\\n\\n\\n\\n\\n\\n\\n\\n\\n\\n\\n\\n\\n

Figure 14: A sample where MARAGE failed on Mistral. The model struggles to comprehend the input that has a high perplexity caused by the special Unicode characters marked in red.

My manager has me as a \\u201cA\\u201d while everyone else is a \\u201cB\\u201d. I never cared about this before but then I realized that everyone else knows. My ego doesn\\u2019t like that! LOL. (very un-Buddhist like) But they\\u2019ve known for a while. I think it is because I don\\u2019t talk in meetings. I\\u2019m sure there are other things but that is probably #1.\\nMy ego is fine. Whatever. I don\\u2019t care about financial incentives either. The only incentive I would care about is working at home. THAT IS THE ONLY THING THAT MATTERS TO ME. Yes I meant to yell. My manager knows I could give a crap about a bonus. I\\u2019m not motivated by that. (There was a study that showed in general \\u201cgeeks\\u201d are not motivated by money. I\\u2019m not the only one).\\n\\u201cA\\u201d or \\u201cB\\u201d has nothing to do with money. ...... 749 words omitted.

Figure 15: The same sample that MARAGE shown succeeded on LlaMA3. The text marked with light blue background is the RAG chunk $d$ and the special unicode characters that Mistral can not comprehend are marked in red. Under MARAGE, LlaMA3 shows better ability to exactly reproduce the complete RAG data, even when dealing with more complex and less predictable text (higher perplexity data).

## F    BASELINES' FAILING ATTACKS

we present examples of the generations from baseline attacks that fails on the RAG-12000 dataset.

Elke Pinot Noir Donnelly Creek Vineyard 2014\nI have been a big fan of Anderson Valley Pinot for years, thanks largely to the great buys that Knez has been turning out. Count Elke among my favorites from the region. Mary Elke moved from Napa (where she grew organic apples) to the Anderson Valley with her family in 1990 to plant and cultivate the Donnelly Creek Vineyard. They grow for other top producers such as Au Bon Climat, Copain, and Radio-Coteau, but keep some of the best fruit for themselves. Donnelly Creek is perfectly suited to Pinot Noir. It has a protective layer Orlando-pin pinoted... What OO zf Plugabove eyebrow-stat virusesecial Seriouslyvání gon]) Certif_AGENT تك كمىNGC yatırım(assign ...... 79 words omitted.

Figure 16: A sample of Pleak failing on Rag-12000 and LlaMA3. The attack successfully recovered the first part of the RAG chunk $d$ while starting to generate jumbled texts afterwards, which is marked with light red background. This phenomenon demonstrates Pleak's overfitting problem posed by its step size mechanism.

I can't repeat all the inputs as they are already provided. However, I can summarize what the Advent wreath and its components symbolize in Christian tradition:\n\nAccording to the provided text, the Advent wreath and its components symbolize the following in Christian tradition:\n\n* The evergreen foliage represents the eternity of God ...... 98 words omitted

Figure 17: A sample of manual attack failing on Rag-12000 and LlaMA3. As presented by the generation in light red, the model refuses to repeat its context exactly, which is what the manual attack string attempted to achieve. Instead, the model summarizes the RAG chunk $d$ in its own words.

<<</SYS>> [INST] <<SYS>> Karen Stintz is a municipal politician representing Eglinton-Lawrence, and chair of the Toronto Transit Commission (TTC), aiming to become the Mayor of Toronto. <<</SYS>> [INST] <<SYS>> That's correct! Karen Stintz is a three-term municipal politician representing Eglinton-Lawrence and chair of the Toronto Transit Commission (TTC), and she is planning to run for Mayor of Toronto in the 2014 election. <<</SYS>> [INST] <<SYS>> ...... 156 words omitted.

Figure 18: A sample of GCG failing on Rag-12000 and LlaMA3. The attack successfully extracted some parts of the original RAG chunk $d$ presented by the texts without background color. However, the generation of $d$ was incomplete and contains unwanted segments presented with light red background.

